# Mitochondria Bioenergetic Functions and Cell Metabolism Are Modulated by the Bergamot Polyphenolic Fraction

**DOI:** 10.3390/cells11091401

**Published:** 2022-04-20

**Authors:** Cristina Algieri, Chiara Bernardini, Francesca Oppedisano, Debora La Mantia, Fabiana Trombetti, Ernesto Palma, Monica Forni, Vincenzo Mollace, Giovanni Romeo, Salvatore Nesci

**Affiliations:** 1Department of Veterinary Medical Sciences, University of Bologna, 40064 Ozzano Emilia, Italy; cristina.algieri2@unibo.it (C.A.); chiara.bernardini5@unibo.it (C.B.); debora.lamantia2@unibo.it (D.L.M.); fabiana.trombetti@unibo.it (F.T.); monica.forni@unibo.it (M.F.); salvatore.nesci@unibo.it (S.N.); 2Department of Health Sciences, Institute of Research for Food Safety & Health (IRC-FSH), University Magna Graecia of Catanzaro, 88100 Catanzaro, Italy; palma@unicz.it; 3Health Sciences and Technologies-Interdepartmental Center for Industrial Research (CIRI-SDV), Alma Mater Studiorum, University of Bologna, 40126 Bologna, Italy; 4Department Gynecological, Obstetrical and Pediatric Sciences, Medical Genetics Unit, Sant’Orsola-Malpighi University Hospital, 40126 Bologna, Italy; egf.giovanni.romeo@gmail.com

**Keywords:** mitochondria, cell metabolism, F_1_F_O_-ATPase, mitochondrial permeability transition pore, porcine aortic endothelial cells, bergamot polyphenolic fraction

## Abstract

The bergamot polyphenolic fraction (BPF) was evaluated in the F_1_F_O_-ATPase activity of swine heart mitochondria. In the presence of a concentration higher than 50 µg/mL BPF, the ATPase activity of F_1_F_O_-ATPase, dependent on the natural cofactor Mg^2+^, increased by 15%, whereas the enzyme activity in the presence of Ca^2+^ was inhibited by 10%. By considering this opposite BPF effect, the F_1_F_O_-ATPase activity involved in providing ATP synthesis in oxidative phosphorylation and triggering mitochondrial permeability transition pore (mPTP) formation has been evaluated. The BPF improved the catalytic coupling of oxidative phosphorylation in the presence of a substrate at the first phosphorylation site, boosting the respiratory control ratios (state 3/state 4) by 25% and 85% with 50 µg/mL and 100 µg/mL BPF, respectively. Conversely, the substrate at the second phosphorylation site led to the improvement of the state 3/state 4 ratios by 15% only with 100 µg/mL BPF. Moreover, the BPF carried out its beneficial effect on the mPTP phenomenon by desensitizing the pore opening. The acute effect of the BPF on the metabolism of porcine aortica endothelial cells (pAECs) showed an ATP rate index greater than one, which points out a prevailing mitochondrial oxidative metabolism with respect to the glycolytic pathway, and this ratio rose by about three times with 100 µg/mL BPF. Consistently, the mitochondrial ATP turnover, in addition to the basal and maximal respiration, were higher in the presence of the BPF than in the controls, and the MTT test revealed an increase in cell viability with a BPF concentration above 200 µg/mL. Therefore, the molecule mixture of the BPF aims to ensure good performance of the mitochondrial bioenergetic parameters.

## 1. Introduction

F-type ATP synthase (F_1_F_O_-ATPase) belongs to the rotary ATPase family and is present in eukaryotic mitochondria, where it functions both as a complex for ATP synthesis and as an ion pump [1,2,3,4]. In particular, F_1_F_O_-ATPase consists of the catalytic F_1_ hydrophilic domain and the F_O_ transmembrane hydrophobic domain [1,5]. The latter is attributed to both a functional and a structural role. In oxidative phosphorylation (OXPHOS), it is involved in the synthesis process of ATP, and the supramolecular organization in dimers arranged in a long row is involved in the formation of mitochondrial *cristae* [6]. As for the synthesis of ATP, it occurs in the mitochondria, starting from ADP and Pi. The energy required for this reaction derives from the proton motive force (Δ*p*) generated by the mitochondrial respiratory complexes [7]. In recent years, it has been shown that the F_O_ transmembrane domain is directly involved in the formation of the mitochondrial permeability transition pore (mPTP), located in the inner mitochondrial membrane (IMM) [8,9,10]. In particular, F_1_F_O_-ATPase, usually activated by the natural cofactor Mg^2+^, is activated by Ca^2+^, undergoing a conformational change [11]. Therefore, mPTP seems to be generated by an increase in the concentration of Ca^2+^, which by opening the pore determines the free passage of ions and solutes through the IMM, leading to the onset of a series of harmful events at the mitochondrial level that can lead to cell death [12,13]. Therefore, research conducted towards identifying compounds that can interact with Ca^2+^-activated F_1_F_O_-ATPase to delay, or rather, inhibit the formation of mPTP becomes important [13,14]. This would be very important as it has been shown that the dysregulation of the mPTP is involved in the onset of serious diseases, such as cancer, ischemia, and heart, as well as neurodegenerative disorders [15].

The Mediterranean diet (Md), defined as part of the “intangible cultural heritage of humanity,” is related to an improvement in health, especially in regard to the reduced incidence of cancer, diabetes mellitus, and, even more significantly, coronary heart disease in the region of Southern Italy [16]. In particular, the reduced intake of red meat alongside the greater consumption of olive oil and plant-based foods means that Md positively affects the onset and progression of heart disease [16]. In this context, bergamot (*Citrus bergamia Risso et Poiteau*) is inserted, a plant endemic to the Calabrian Ionian coast in Southern Italy, a place where the soil and microclimate give the bergamot fruit its peculiar organoleptic characteristics [16,17]. To date, bergamot, in addition to being used in the cosmetic and confectionery industries, is used in nutraceutical supplementation in many pathologies, thanks to its high concentration of polyphenols. In particular, bergamot is rich in flavonoid glycosides, such as neoeriocitrin, neohesperidin, naringin, and glycosylated polyphenols, such as bruteridin and melitidin [18]. From the juice and albedo of bergamot the BPF is obtained, which is the polyphenol-rich fraction, with antioxidant, anti-inflammatory, lipid-lowering, and hypoglycemic effects [16]. As a consequence of these characteristics, the BPF seems to have a beneficial role in the pathophysiological mechanisms of diabetic cardiomyopathy, in particular in mitochondrial and sarcoplasmic dysfunctions. In fact, it is known that diabetic cardiomyopathy is characterized by mitochondrial oxidative stress and calcium alteration, caused by sarcoplasmic reticulum dysfunction, conditions that lead to cardiomyocyte death [19]. Therefore, the BPF, with its high naringin content, could protect against sarcoplasmic reticulum stress. These effects of the BPF were demonstrated in an in vitro model of cardiotoxicity. This condition was induced in rat embryonic cardiomyoblasts (H9c2) treated with doxorubicin, and the role of mitochondria and the endoplasmic reticulum was examined. The data obtained showed that the BPF reduced oxidative damage and cell death. Furthermore, a higher lipid content and a reduced escape of calcium ions from the reticulum have been reported [20]. Furthermore, the BPF could be added to drug therapy in the treatment of patients with cancer [21]. In rats, the BPF has been shown to reduce chemotherapy-induced peripheral neuropathy, acting on mechanical allodynia and thermal hyperalgesia, both of which are determined by paclitaxel treatment. In addition to this, the BPF was used in a clinical trial concerning schizophrenia. Pharmacological therapy supplementation with the BPF showed an improvement in the cognitive executive functioning of the patients, probably linked to the neuroprotective activity (antioxidant and antiapoptotic) of the flavonoids naringin and neohesperidin present in the BPF [22]. Following these characteristics and the involvement of the BPF in all of these pathological conditions, we decided to study its eventual action at the level of F_1_F_O_-ATPase and of the mPTP formation as well as the subsequent effect on the cell metabolism.

## 2. Materials and Methods

### 2.1. Chemicals and Reagents

Oligomycin (a mixture of oligomycins A, B, and C) and Fura-FF were purchased from Vinci-Biochem (Vinci, Italy). Na_2_ATP, rotenone, and antimycin A were obtained from Sigma–Aldrich (Milan, Italy) and the BPF from the Herbal & Antioxidant Derivatives (H&AD) company (Bianco, Italy). Seahorse XF Assay Kits and Reagents were purchased from Agilent, Santa Clara, CA, USA. Quartz double-distilled water was used for all reagent solutions.

### 2.2. Preparation of the Bergamot Polyphenolic Fraction (BPF)

The harvest of *C. bergamia Risso & Poiteau* fruits took place in a geographical area of about 90 km between Bianco and Reggio Calabria, Italy. The fruits were peeled and squeezed to obtain the bergamot juice. Subsequently, the juice was oil fraction depleted by stripping and, using an ultrafiltration process, clarified. After this step, the clarified juice was passed through a polystyrene resin column that absorbed polyphenol compounds with a molecular weight between 300 and 600 Da. The elution of the polyphenol fractions occurred using a KOH solution. The reduction in furocoumarin content was achieved by incubating the basic eluate on a rocking platform and adjusting the shaking time according to the amount of furocoumarin contaminants. The phytocomplex, thus obtained, was filtered on cationic resin at an acidic pH to be neutralized. Lastly, the BPF powder was obtained after the vacuum drying and mincing of the neutralized phytocomplex. Furthermore, the content of flavonoids, furocoumarins, and other polyphenols present in the BPF powder was determined. In particular, the UHPLC-HRMS/MS method made it possible to establish that the BPF comprises more than 40% flavonoids and about 60% carbohydrates, fatty acids, pectins, and maltodextrins. The flavonoid profile, identified by high-resolution mass spectrometry (Orbitrap spectrometer) and confirmed by HRMSMS (ddMS2, data-dependent MS/MS), consists of neoeriocitrin, naringin, and neohesperidin, and the whole HMG family, including bruteridin and melitidin, and flavonoids, such as 6.8-di-C-glycosides [18,23].

### 2.3. Preparation of the Mitochondrial Fractions

Immediately after slaughter, the hearts of adult swine (*Sus scrofa domesticus*) were isolated, stored on ice at 0–4 °C in a local abattoir, and transferred to the laboratory over the next 2 h. At this point, after eliminating fat and blood clots, about 30–40 g of heart tissue was removed and washed several times with an ice-cold Tris-HCl buffer (0.25 M sucrose/10 mM Tris (hydroxymethyl)-aminomethane (Tris), pH 7.4) (medium A). A preparation was obtained for each heart, in which the tissue was cut with scissors, dried with paper, and weighed. Each gram of fresh-chopped tissue was homogenized in 10 mL of a buffer (medium B) consisting of 0.25 M sucrose, 10 mM Tris (pH 7.4 with HCl), 1.0 mM EDTA (free acid), and 0.5 mg/mL fatty-acid-free bovine serum albumin (BSA). The tissue was subjected to a first break with Ultraturrax T25 and, subsequently, homogenized with a motor-driven Teflon pestle homogenizer (Braun Mel-sungen Type 853202). Homogenization was performed at 650 rpm three times. The homogenate thus obtained was subjected to stepwise centrifugation (Sorvall RC2-B, rotor SS34). In fact, after the first centrifugation at 1000× *g* for 5 min, the obtained pellet was resuspended and homogenized again in the previous conditions and recentrifuged at 1000× *g* for 5 min. The supernatants obtained following the two centrifugation cycles were combined, filtered through 4 layers of cotton gauze, and centrifuged at 10,500× *g* for 10 min. The result of this centrifugation is the raw mitochondrial pellet, which, after being resuspended in medium A, was recentrifuged at 10,500× *g* for 10 min. The pellet obtained following this last centrifugation is constituted by the final mitochondrial fraction [24]. This pellet was resuspended in a small volume of medium A and subjected to gentle agitation with a Teflon Potter Elvejehm homogenizer. The samples were always kept at a temperature between 0 and 4 °C. The colorimetric method of Bradford by the Bio-Rad Protein Assay kit II, using BSA as the standard, allowed for the evaluation of the protein concentration of the samples, which was found to be equal to 30 mg/mL. Finally, the mitochondrial preparations thus obtained were stored in liquid nitrogen and subsequently used for the studies conducted on the F-ATPase activity [25].

### 2.4. Mitochondrial F-ATPase Activity Assays

Immediately after thawing the mitochondria stored in liquid nitrogen, mitochondrial preparations were used to evaluate the F-ATPase activity. To measure the ATP hydrolysis capability by the Mg^2+^-activated F_1_F_O_-ATPase, 1 mL of a reaction medium consisting of 0.15 mg of mitochondrial protein and 75 mM ethanolamine-HCl buffer at pH 9.0 was used in the presence of 6.0 mM Na_2_ATP and 2.0 mM MgCl_2_, while to determine the activity of Ca^2+^-activated F_1_F_O_-ATPase the same buffer was used but at pH 8.8 in the presence of 3.0 mM Na_2_ATP and 2.0 mM CaCl_2_. The assay involves a preincubation of 5 min at 37 °C. Subsequently, maintaining the same temperature, the Na_2_ATP substrate was added to start the reaction. After 5 min, the reaction was stopped using 1 mL of an ice-cold 15% (*w*/*w*) trichloroacetic acid (TCA) aqueous solution. At this point, the samples were centrifuged for 15 min at 3500 rpm (Eppendorf Centrifuge 5202). To indirectly determine the F-ATPase activity, the concentration of inorganic phosphate (Pi) hydrolyzed by known amounts of mitochondrial protein present in the supernatant was defined spectrophotometrically. Therefore, before the reaction started, 1.0 μL of 3.0 mg/mL oligomycin in dimethylsulfoxide (DMSO) was added to the mixture. For each series of experiments, at the same time as the conditions to be tested, the total ATPase activity was calculated by evaluating the Pi in control tubes containing 1.0 μL of DMSO per mL reaction system. To date, oligomycin is used in F-ATPase assays as it represents a specific inhibitor of F-ATPase capable of selectively blocking the F_O_ subunit. In the experiments we conducted, a 3.0 mg/mL dose of oligomycin gave the maximum inhibition of F-ATPase. In each experiment, the F_1_F_O_-ATPase activity was obtained as the difference between the Pi hydrolyzed in the presence of oligomycin and the Pi hydrolyzed by the total ATPase activity and expressed as μmol Pi·mg protein^−1^·min^–1^ [26]. The concentration of inorganic phosphate (Pi) hydrolyzed by known amounts of mitochondrial protein, which is an indirect measure of ATPase activity, was spectrophotometrically evaluated according to Fiske and Subbarow [27].

### 2.5. Evaluation of Oxidative Phosphorylation

Immediately after the fresh preparation of the swine heart mitochondrial fraction, the mitochondrial respiratory activity was polarographically evaluated by a Clark-type electrode using a thermostated Oxytherm System (Hansatech Instruments, King’s Lynn, UK) in terms of oxygen consumption at 37 °C in a 1 mL polarographic chamber. The reaction mixture, maintained at a fixed temperature and continuously stirred, contained a 0.25 mg/mL mitochondrial suspension, 40 mM KCl, 0.2 mg/mL fatty-acid-free BSA, 75 mM sucrose, 0.5 mM EDTA, 30 mM Tris–HCl, pH 7.4, and 5 mM KH_2_PO_4_, plus 3 mM MgCl_2_. The reaction was carried out at a fixed temperature and under constant stirring. The assessment of the rate of oxygen consumption was made in the presence of specific substrates. In regard to complex I, the substrates were glutamate/malate in a ratio of 1:1; for complex II, the substrate was succinate. Furthermore, complex I was inhibited by 1 μg/mL of rotenone, while the inhibition of complex III was achieved by 1 μM of antimycin A. The oxidation of glutamate/malate made it possible to determine the activity of NADH; for ubiquinone oxidoreductase, on the other hand, the oxidation of succinate represented the multicomponent succinoxidase pathway, i.e., the electron flux in the respiratory chain through complex II. By adding 1 µM of antimycin A to the respective tests with glutamate/malate and succinate, it was possible to evaluate the nonspecific oxygen consumption. To evaluate the coupling between respiratory activity and phosphorylation, 150 nmol ADP at state 2 of respiring mitochondria was added [28,29]. The respiratory control ratio (RCR) of OXPHOS was assessed as the ratio between state 3 (when ATP is synthesized) and state 4 (when ATP is not synthesized). To evaluate the effects of the BPF, the mitochondrial suspensions were added at the same time as 50 µg/mL of the BPF or 100 µg/mL of the BPF solutions to the polarographic chamber before starting the reaction at 37 °C. Respiratory activities were expressed as nmoles O_2_∙mg protein^−1^∙min^−1^. The experimental protocol provided for the injection (in a very precise order) of reagents into the polarographic cell using a syringe. In particular, to mitochondrial protein suspensions, in the presence and absence of the BPF, the following were added: inhibitors of the previous respiratory chain steps (when required), substrate(s), ADP, and inhibitors (rotenone for glutamate/malate-stimulated respiration and antimycin A for succinate-stimulated respiration) [30]. The rate of oxygen consumption was assessed in the presence of the specific substrates, glutamate/malate for complex I and succinate for complex II. Polarographic assays were performed at least in triplicate on three mitochondrial preparations from different animals.

### 2.6. MPTP Evaluation

Immediately after the fresh preparation of swine heart mitochondrial fractions, fresh mitochondrial suspensions (1 mg/mL) were energized in the assay buffer (130 mM KCl, 1 mM KH_2_PO_4_, 20 mM HEPES, and pH 7.2 with TRIS), incubated at 37 °C with 1 μg/mL rotenone and 5 mM succinate. The effect of the BPF was tested by adding BPF concentrations of 0.01 µg/mL or 50 µg/mL to mitochondrial suspensions before evaluating the mPTP. In regard to the induction of the mPTP opening, this was determined by a CaCl_2_ solution added at intervals of 1 min to guarantee the presence of low Ca^2+^ concentrations equal to 10 μM. Spectrofluorophotometry analysis, carried out in the presence of 0.8 μM Fura-FF, allowed for the evaluation of the calcium retention capacity (CRC), the reduction of which indicates the mPTP opening. The probe used has different spectral characteristics depending on the Ca^2+^ absence or presence. In fact, in the absence of Ca^2+^ (Fura-FF low Ca^2+^), an excitation/emission spectrum at 365/514 nm is detected; alternatively, when the concentration of Ca^2+^ increases (Fura-FF high Ca^2+^), the spectrum shifts to 339/507 nm. The increase in the fluorescence intensity ratio (Fura-FF high Ca^2+^)/(Fura-FF low Ca^2+^) is determined by the reduction in CRC and, therefore, allows for the evaluation of the opening of the mPTP [31]. The LabSolutions RF software was used to process these measurements.

### 2.7. Cell Cultures

Primary cell cultures of porcine aortic endothelial cells (pAECs) were isolated and maintained as previously described [32]. pAECs from 3 to 8 passages (P) were used to perform the experiments. The cells were seeded and routinely cultured in T25 or T75 primary culture flasks (2 × 10^4^ cells/cm^2^) in a human endothelial serum-free medium (hESFM), added with 5% FBS and 1× antibiotic/antimycotic solution in a 5% CO_2_ atmosphere and 38.5 °C. An inverted Eclipse Microscope (TS100) with a digital C-Mount Nikon photo camera (TP3100) was used to define the cell morphology.

### 2.8. Cellular Metabolism

The measurement of the oxygen consumption rate (OCR) and the index of cell respiration (pmol/min), as well as the extracellular acidification rate (ECAR) and the index of glycolysis (mpH/min), was carried out simultaneously using the Seahorse XFp analyzer (Agilent, USA).

pAECs (20 × 10^3^/well) were grown in XFp cell culture miniplates (Agilent, USA) for 24 h. On the day of the experiment, cells were switched to a freshly made Seahorse XF DMEM medium of pH 7.4 supplemented with 10 mM glucose, 1 mM sodium pyruvate, and 2 mM L-glutamine. Analyses were conducted in the absence (control) and in the presence of 50 μg/mL, 100 μg/mL, and 150 μg/mL of the BPF for the ATP Rate Assay and 100 μg/mL of BPF for the Mito Stress Test. The OCR and ECAR were measured with the ATP Rate Assay, Cell Mito Stress Test, and Glycolysis Stress Test programs after the plates were incubated for 45 min at 37 °C in air. In addition, the injection ports of the XFp sensor cartridges were hydrated overnight with XF calibrant at 37 °C; they were subsequently loaded with 10× the concentration of inhibitors, as indicated by the instructions for the Seahorse XFp ATP Rate Assay, Cell Mito Stress Test, and Glycolysis Stress Test. Final concentrations of 1.5 μM oligomycin (port A) and 0.5 μM rotenone (Rot) plus 0.5 μM antimycin A (AA) (port B) were used for the ATP Rate Assay. Instead, for the Cell Mito Stress Test, the final concentrations were 1.5 μM oligomycin (olig) (port A), 1.0 μM carbonyl-cyanide-4-(trifluoromethoxy) phenylhydrazone (FCCP) (port B), and 0.5 μM rotenone plus antimycin A (port C) [33]. Finally, for the Glycolysis Stress Test the concentrations used were 10 mM glucose (port A), 1 μM oligomycin (port B), and 50 mM 2-deoxyglucose (2DG) (port C).

The ATP Rate Assay provides the bioenergetic parameters currently used to characterize the cellular ATP production, namely ATP production rate, related to the conversion of glucose to lactate in the glycolytic pathway (glycoATP production rate) and to mitochondrial OXPHOS (mitoATP production rate). Accordingly, the ratio between the mitoATP production rate and the glycoATP production rate (ATP rate index) is currently considered a valuable parameter to detect changes and differences in the metabolic phenotype (a ratio > 1 means mainly OXPHOS pathway; ratio < 1 means mainly glycolytic pathway).

The Mito Stress Test enables the characterization of cell respiration via the following parameters: basal respiration, detected as the baseline OCR before oligomycin addition; minimal respiration, measured as the OCR in the presence of oligomycin; and maximal respiration, evaluated as the OCR after FCCP addition. The so-called proton leak, which corresponds to the difference between the basal respiration and the respiration in the presence of oligomycin (minimal respiration), indicates the re-entry of H^+^ in the intermembrane space independently of the ATP synthase. The nonmitochondrial respiration, evaluated as the OCR in the presence of rotenone plus antimycin A (respiratory chain inhibitors), was subtracted from all the above parameters. The ATP turnover, or oligomycin-sensitive respiration, was obtained from the difference between the basal respiration and the minimal respiration (OCR in the presence of oligomycin). Finally, the difference between the maximal and the basal respiration provided the spare capacity, which represents the ability to respond to increased energy demand and can be considered a measure of the flexibility of the OxPhos machinery.

The Glycolysis Stress Test is the assay for measuring the glycolytic function in cells. The Agilent Seahorse XFp evaluates the ECAR to assess the key parameters of glycolytic flux: glycolysis, glycolytic capacity, and glycolytic reserve. The ECAR, before glucose injection, when cells are incubated in the glycolysis stress test medium with 2 mM glutamine without glucose or pyruvate, is not dependent on glycolysis. Cells after the addition of saturating amounts of glucose have a response, reported as the rate of glycolysis under basal conditions. The addition of oligomycin shifts the energy production to glycolysis, with the subsequent increase in ECAR revealing the cellular maximum glycolytic capacity. The difference between the glycolytic capacity and glycolysis rate defines the glycolytic reserve. The addition of 2DG measures other sources of extracellular acidification that are not attributed to glycolysis and is referred to as nonglycolytic acidification.

The analyses were carried out at a temperature of 37 °C, and the data were analyzed using the WAVE software, normalizing the OCR and ECAR values with respect to the total number of cells per well. The values for the various parameters were measured per well, following the manufacturer’s instructions. The ATP Rate Assay, the Mito Stress Test, and the Glycolysis Stress Test were performed three times in independent experiments.

### 2.9. Cell Viability

To investigate the effect of the BPF on cell viability, pAECs were seeded in 96-well plates at a density of 1 × 10^4^ cells/well, and, the day after, were exposed to increasing doses of the bergamot polyphenolic fraction (BPF) (1, 5, 10, 25, 50, 100, 150, 200, 250, and 300 µg/mL) for 5 h. The viability was determined using the MTT assay. Briefly, the addition of the culture medium with the MTT substrate was followed by a three-hour incubation. In addition, the MTT solubilization solution allowed for the dissolving of the formazan crystals. The Infinite^®^ F50/Robotic absorbance microplate readers from TECAN Life Sciences(Männedorf, Switzerland) made it possible to measure the formazan absorbance (Abs) at a wavelength of 570 nm, from which the background absorbance was subtracted from multiwell plates, measured at 690 nm.

### 2.10. Statistical Analysis

Three independent experiments were performed for each treatment, which were replicated 6 or 8 times (viability test). Data were analyzed with GraphPad PRISM 7.0 (GraphPad Software, Inc., La Jolla, CA, USA). Results are shown as mean ± SD. Data were analyzed by a one-way ANOVA followed by a Tukey’s test. A *p*-value ≤ 0.05 was considered statistically significant.

## 3. Results

The effect of the BPF on mitochondria bioenergetics has been evaluated. The experiments considered the acute effect of the BPF on isolated mitochondria and pAECs’ metabolism. We tried to exploit the BPF mechanism at the molecular level to understand its modulatory role in the Ca^2+^-activated F_1_F_O_-ATPase when the mPTP forms, as well as in the cell energy metabolism.

### 3.1. Effect of the BPF on Isolated Mitochondria

#### 3.1.1. Mg^2+^- and Ca^2+^-activated F_1_F_O_-ATPase Activity in the Presence of the BPF

In the range of 0.01–100 µg/mL BPF, the effect was evaluated on Mg^2+^- and Ca^2+^-activated F_1_F_O_-ATPase (Figure 1). F_1_F_O_-ATPase sustains ATP hydrolysis in a divalent cofactor-dependent way. Increasing BPF concentrations had an opposite effect on the enzyme activity. In Mg^2+^-activated F_1_F_O_-ATPase, an increase in activity was induced by the BPF. Conversely, in Ca^2+^-activated F_1_F_O_-ATPase, an inhibitor effect was detected in the presence of the BPF. However, the main difference between the F_1_F_O_-ATPase activities is that the enzyme in the presence of Mg^2+^ attained 15% activation at concentrations higher than 50 µg/mL BPF, whereas a constant 10% of inhibition at BPF concentrations below 50 µg/mL was detected by Ca^2+^ (Figure 1).

#### 3.1.2. BPF Effect on Oxidative Phosphorylation

To understand the relationship between substrate oxidation by mitochondrial complexes and the BPF stimulation of Mg^2+^-activated F_1_F_O_-ATPase, the mitochondrial respiration coupled to ATP synthesis was evaluated in the presence of glutamate/malate and succinate as substrates for the first and second phosphorylation sites, respectively. Preliminary tests assessed that, under these conditions, O_2_ consumption was suppressed by 1 mM KCN, a known inhibitor of cytochrome *c* oxidase, or complex IV.

The results obtained when different concentrations of the BPF were added to state 3 respiring (ADP-stimulated) mitochondria showed no significant difference with respect to the control (absence of the BPF) in oxygen consumption, both in the presence of NAD-dependent substrates (first phosphorylation site) (Figure 2A) and in the presence of succinate (second phosphorylation site) (Figure 2B). Likewise, the respiration that occurred in state respiratory activity, which mirrors the slowdown in oxygen consumption when the added ADP is consumed, being phosphorylated to ATP, was unaffected by the BPF independently of the substrate to the first or second phosphorylation site (Figure 2A,B). However, the coupling between substrate oxidation (glutamate/malate or succinate) and ADP phosphorylation, evaluated as the state 3/state 4 ratio, was improved by the BPF. The RCR value obtained in regard to the first phosphorylation site increased by 25% or 85% with 50 µg/mL and 100 µg/mL BPF, respectively (Figure 2A). The state 3/state 4 ratio increased the RCR value at the second phosphorylation site by 15% only in the presence of 100 µg/mL BPF (Figure 2B).

#### 3.1.3. MPTP Desensitization to Ca^2+^

The mPTP opening is measured when the Ca^2+^ pulses accumulated in the mitochondrial matrix were released. An increase in the Ca^2+^ concentration in the mitochondria stimulates mPTP opening, and the capability of the mitochondria to accumulate Ca^2+^ concentration without mPTP formation is measured as the calcium retention capacity (CRC). On these bases, the CRC was evaluated by adding 10  μM Ca^2+^ at subsequent steps of 1 min to succinate-energized freshly prepared mitochondrial suspensions. The CRC decrease, which is detected by increasing the Fura-FF ratio intensity, reveals the mPTP opening. Accordingly, the increase in the CRC upon subsequent 10 μM Ca^2+^ additions at fixed time intervals indicated that, in the presence of the BPF, mitochondria attained a higher threshold value of Ca^2+^ concentration than the control (absence of the BPF) to trigger mPTP formation (Figure 3).

Importantly, the two concentrations of the BPF tested (50 μg/mL and 100 μg/mL) did not have different desensitizing powers on the mPTP. However, 50 μg/mL BPF attained a lower CRC value (high Fura-FF ratio) than 100 μg/mL BPF at the mPTP opening. On these bases, a lower CRC value would mirror a larger mPTP size.

### 3.2. Effect of BPF on pAECs

#### BPF Effect on pAECs’ Cellular Metabolism and Viability

The production of ATP by OXPHOS or glycolysis at different BPF concentrations (0, 50, 100, and 150 µg/mL) is shown in Figure 4 by OCR and ECAR values, respectively, under basal metabolic conditions. The sum of the mitoATP and glycoATP production rates, which is the total ATP production by pEACs, was unaffected by the BPF. However, mitoATP production was increased by 20%, whereas that of glycoATP was reduced by 34% in the presence of 100 µg/mL BPF (Figure 4A). Accordingly, the ratio between the mitoATP production rate and the glycoATP production rate, known as the ATP rate index, in the presence of 100 µg/mL BPF was approximately two times greater than the control (Figure 4B).

To verify the metabolism remodeling of pEACs by the BPF, analyses of mitochondrial bioenergetics and the glycolysis pathway have been performed (Figure 5). The cell respiration profile of cells treated with and without BPF was shown in Figure 5A, and we did not notice any difference in the responses to mitochondrial inhibitors. Nonetheless, the key parameters of mitochondrial activities, known as basal respiration, were detected as the baseline OCR before the addition of oligomycin; the minimal respiration, measured as the OCR in the presence of oligomycin that corresponds to the proton leak; the maximal respiration, evaluated as the OCR after the addition of FCCP; spare respiratory capacity provided from the difference between the maximal and basal respiration; and ATP turnover, as the oligomycin-sensitive OCR has been calculated (Figure 5B). All of the parameters, except the spare respiratory capacity, were increased in the presence of the BPF. The improvement of cell respiration at basal or under-stimulation (maximal respiration) by the BPF with respect to the control was coupled with an increase in the ATP turnover. The glycolytic profile in the presence of the BPF showed an ECAR activity lower than that of the control (Figure 5C). Consequently, we detected a reduction of 20% of the glycolytic parameters in the presence of the BPF, i.e., glycolysis and the glycolytic capacity. As the difference between the glycolytic capacity and the glycolysis rate defines the glycolytic reserve, and the BPF induced the same inhibiting power on them, there was no effect of the BPF on the glycolytic reserve (Figure 5D).

The effect of increasing doses of the BPF on pAECs’ viability was evaluated. The MTT results showed no toxicity in all of the BPF concentrations tested (Figure 6). Conversely, a significant increase in cell viability was evident at higher doses of the BPF (200, 250, and 300 µg/mL).

## 4. Discussion

Mitochondrial dysfunction arises from the alteration in energy homeostasis by impaired OXPHOS function [34,35]. The decline in ATP production in cells associated with increased ROS production triggers molecular events responsible for mPTP formation [36]. In the OXPHOS system, F_1_F_O_-ATPase can switch from the energy-producing to the energy-dissipating molecular machine. The dual role of F_1_F_O_-ATPase is dependent on the cation cofactor that sustains the enzyme activity. The natural cofactor Mg^2+^, bound to the catalytic sites, provides the bifunctional role of ATP synthesis and hydrolysis coupled to H^+^ translocation. Otherwise, the substitution of Mg^2+^ in the catalytic sites with Ca^2+^, when its concentration suddenly increases in the mitochondria, establishes the monofunctional feature of ATP hydrolysis that drives the H^+^ pump [37]. In all likelihood, Ca^2+^-activated F_1_F_O_-ATPase might be the molecular component that supports the properties for the formation of mPTP [38,39,40]. The BPF has an opposite effect on Mg^2+^- and Ca^2+^-activated F_1_F_O_-ATPase. The first is activated, whereas the latter is inhibited. Consistently, we encounter the same physiological effect on the OXPHOS activity and the mPTP opening, respectively. The natural cofactor Mg^2+^, which sustains ATP synthesis in the presence of the BPF, improves the respiratory control ratios (state 3/state 4), a measurement of the coupling between substrate oxidation and phosphorylation [41]. The mitochondria perform the chemiosmotic process of energy transduction better, maximizing the use of the protonmotive force to drive ATP synthesis.

The mitochondria perform higher ATP synthesis stimulated by the BPF. Consistently, Mg^2+^-activated F_1_F_O_-ATPase can benefit from the polyphenolic phytochemicals present in the BPF by noncovalent interactions with the hydrophilic F_1_ domain of the enzyme [42]. The flavones of the BFP have similar structures to resveratrol and quercetin, which only bind to the F_1_ domain and the enzyme–flavone interaction, inhibiting ATP hydrolysis but not ATP synthesis [43]. The common binding site of the polyphenolic compounds lie inside the annulus of the αβ trimer, and these interact with the C-terminal tip of the γ subunit and the β_TP_ subunit conformation as there is no hydrophobic pocket between the γ subunit and either the β_DP_ or β_E_ subunit conformations needed to accommodate the compounds. The improvement of the bifunctional F-ATPase activity in the presence of Mg^2+^, i.e., the ATP synthesis or hydrolysis, probably stems from an additive or synergic action of different compounds present in the BPF on the mitochondrial bioenergetics.

Conversely, the inhibitor effect of the BPF on Ca^2+^-activated F_1_F_O_-ATPase, by corroborating the negative effect, concentration-dependent on the opening and size of the mPTP, highlights the protective effect of the BPF on mitochondrial dysfunction [17]. The Ca^2+^-activated F_1_F_O_-ATPase activity is linked to the mPTP formation and opening [30,31,42,44,45,46]. In the presence of the BPF, the different conformations of the F_1_F_O_-ATPase, when the natural cofactor Mg^2+^ is substituted with Ca^2+^ [8], might make the enzyme unable to induce the profound structural changes transmitted from the hydrophilic F_1_ domain to the membrane needed to form the pore in the F_O_ domain [11,47]. The results are addressed to focus on the aspect of Ca^2+^ signaling within the F_1_F_O_-ATPase, as Ca^2+^ is indispensable for the pore opening, and F_1_F_O_-ATPase is “well-equipped” to transfer the Ca^2+^ signals binding the cation in the catalytic sites [8]. However, it is ascertained that there are changes in the interactions between different divalent cations and the enzyme. Moreover, the amino acid positions of the β subunit are affected by the size of the cofactor [48]. Ca^2+^ has a higher atomic radius than Mg^2+^ and can trigger the signaling transmission of mPTP opening by the modification of the structure of the F_1_ domain [11,49,50].

The BPF action on isolated mitochondria is confirmed by pAECs’ cellular metabolism results. We encounter an effect on ATP production that performs the better mitochondrial activity. The total intracellular ATP levels produced from OXPHOS and glycolysis, the two main metabolic pathways responsible for ATP production in mammalian cells, are not affected by the BPF. However, the positive effect of 100 µg/mL BPF on mitochondria is revealed with an increase in ATP production by oxidative metabolism at the expense of glycolysis. The mitochondrial function could be beneficial for optimal health, whereas mitochondrial dysfunction has long been associated with aging and age-related disease development [50]. The BPF with the polyphenols’ antioxidant properties contributes to the prevention of oxidative stress by increasing mitochondrial biogenesis and ensuring the correct biomolecular redox state, which improves mitochondrial function and energy production [51]. The mitochondrial parameters ATP turnover and basal as well as maximal respiration were significantly ameliorated with the BPF. Contrariwise, the consumption of glucose and the glycolytic capacity of pAECs show a decrease in metabolic activity. In the presence of the BPF, the OXPHOS boosts the substrate oxidation and the cell respiration is exploited for ATP synthesis. However, the BPF does not affect the capability of pAECs to respond to an energetic demand increasing the glycolytic function as no difference in the glycolytic reserve is found with and without the treatment of the cells with the BPF. A metabolic exchange switch could occur when the molecular mechanism regulated by the BPF drives the pAECs towards the aerobic oxidation of the substrates. As pAECs are not cell types with higher energy demands, mitochondria in endothelial cells primarily function in signaling cellular responses to environmental cues [52]. Moreover, mitochondrial content in endothelial cells is relatively modest, and glycolysis sustains part of the bioenergetic cost of cellular functions and events. The metabolic rewiring in pAECs with BPF allows that energy in the form of ATP is continuously replenished by OXHOS in mitochondria and, to a lesser extent, by glycolysis. In all likelihood, the metabolic flux of substrates was addressed by the BPF towards their oxidation in mitochondria, which is controlled by a complex network of enzymatic and signaling pathways, and the chemical composition and the antioxidant proprieties of the BPF influence the overall health of cells.

## 5. Conclusions

In general, the BPF is known to have protective effects on human health by lowering lipids, its antidiabetic action, and by modulating oxidative stress and inflammatory biomarkers [53]. The results obtained also highlight the beneficial effects of the BPF on mitochondrial bioenergetics and oxidative metabolism, counting them as further biological targets for the healthy effect of the BPF on humans.

## Figures and Tables

**Figure 1 cells-11-01401-f001:**
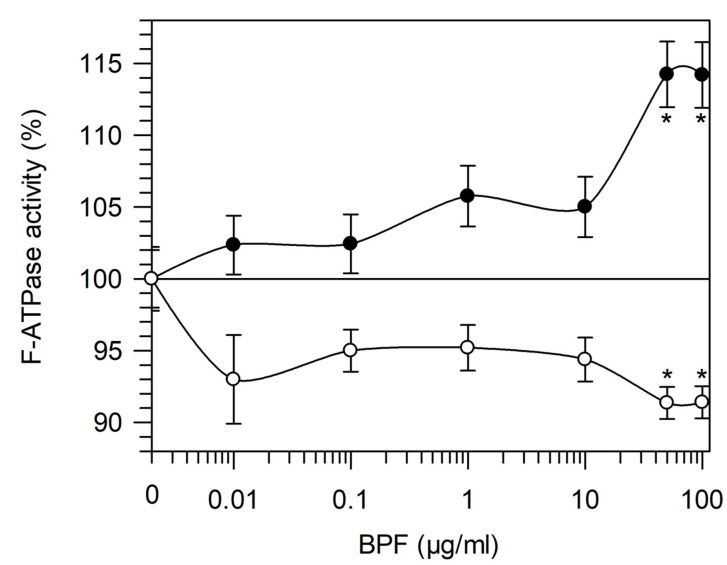
Dose–response curve of the BPF on F_1_F_O_-ATPase activity. F_1_F_O_-ATPase activated by Mg^2+^ (Mg^2+^-activated F_1_F_O_-ATPase) (●) and by Ca^2+^ (Ca^2+^-activated F_1_F_O_-ATPase) (○) activities in the presence of increasing BPF concentrations are expressed as percentages of the total mitochondrial F-ATPase activity sustained by Mg^2+^ or Ca^2+^, respectively. Data represent the mean ± SD from three independent experiments carried out on different mitochondrial preparations. * indicates significant differences with respect to the control (*p*  ≤  0.05).

**Figure 2 cells-11-01401-f002:**
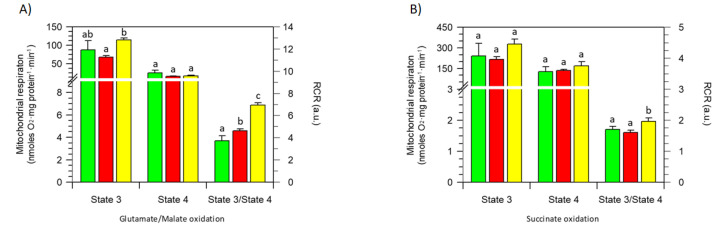
BPF effects on selected oxidative phosphorylation parameters: state 3 and 4 respiration rates and their ratio as the RCR. Glutamate/malate- (**A**) and succinate- (**B**) stimulated mitochondrial respiration in the presence of 50 µg/mL (red) (█), 100 µg/mL (yellow) (█), and in the absence (green) (█) of the BPF, respectively. Data expressed as columns represent the mean ± SD from three independent experiments carried out on different mitochondrial preparations. Different letters indicate significant differences (*p* ≤ 0.05) among treatments within the same parameter (state 3; state 4, and state 3/state 4).

**Figure 3 cells-11-01401-f003:**
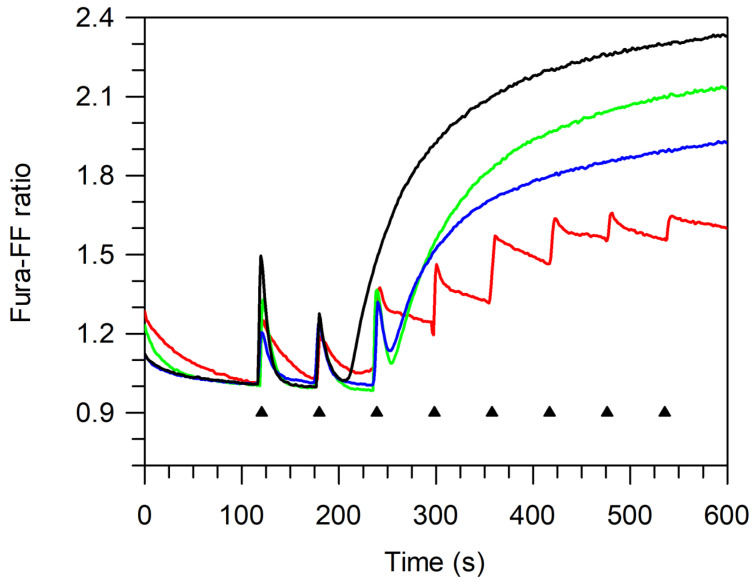
Evaluation of mPTP opening. Representative curves of at least three different experiments on the calcium retention capacity (CRC). The CRC was monitored in response to subsequent 10 μM CaCl_2_ pulses (shown by the triangles), as detailed in Section 2.6, in the absence (control—black line) (**−**) and in the presence of the inhibitors 2 mM MgADP (red line) (**−**), 50 µg/mL BPF (green line) (**−**), and 100 µg/mL BPF (blue line) (**−**).

**Figure 4 cells-11-01401-f004:**
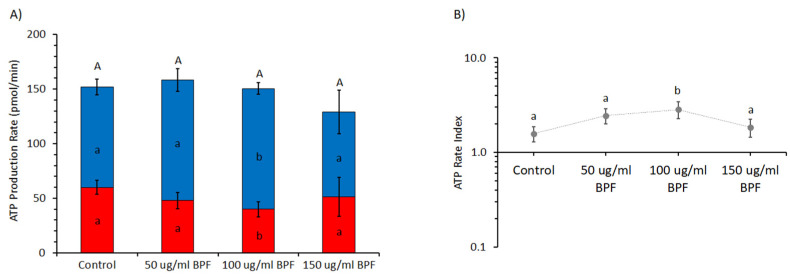
Effect of the BPF on the real-time ATP production rate in pAECs. (**A**) Evaluation of ATP production rate by mitochondrial OXPHOS (blue) (█) or by glycolysis (red) (█) in BPF-treated cells. (**B**) The ATP rate index, calculated as the ratio between the mitochondrial ATP production rate and the glycolytic ATP production rate, is shown on the *y*-axis (logarithmic scale) in pEACs treated without (control) and with the BPF. Data expressed as column charts ((**A**) plots) and points ((**B**) plots) represent the mean ± SD (vertical bars) from three experiments carried out on distinct cell preparations. Different lower-case letters indicate significantly different values (*p* ≤ 0.05) among BPF treatments (0, 50, 100, and 150 μg/mL) in the same metabolic pathway (OXPHOS or glycolysis) (**A**) and ratio value (**B**); different upper-case letters indicate different values (*p* ≤ 0.05) among treatments in ATP production rates due to sum of OXPHOS plus glycolysis (**A**).

**Figure 5 cells-11-01401-f005:**
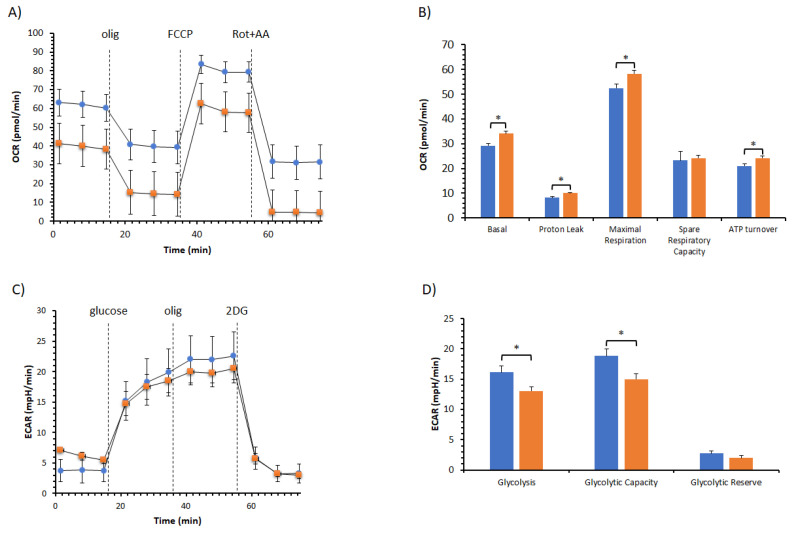
Effect of the BPF on the cell metabolism of pAECs. (**A**) The mitochondrial respiration profile was obtained from the oxygen consumption rate (OCR) without (●, blue) and with 100 μg/mL BPF (■, orange) under basal respiration conditions and after the addition of 1.5 μM oligomycin (olig), 1.0 μM FCCP, and a mixture of 0.5 μM rotenone plus antimycin A (Rot + AA). Inhibitor injections are shown as dotted lines. (**B**) Mitochondrial parameters (basal respiration, ATP production, proton leak, maximal respiration, spare respiratory capacity, and ATP turnover) in the absence (█, blue) or in the presence of 100 μg/mL BPF (█, orange). (**C**) The glycolysis profile was obtained from the extracellular acidification rate (ECAR) without (●, blue) and with 100 μg/mL BPF (■, orange) under basal glycolysis conditions and after the addition of 10 mM glucose (port A), 1 μM oligomycin (port B), and 50 mM 2-deoxyglucose (2DG). Compound injections are shown as dotted lines. (**D**) Glycolytic parameters (glycolysis, glycolytic capacity, and glycolytic reserve) in the absence (█, blue) or in the presence of 100 μg/mL BPF (█, orange). Data expressed as points (**A**,**C**) and column charts (**B**,**D**) represent the mean ± SD (vertical bars) from three experiments carried out on different cell preparations. * indicates significant differences (*p* ≤ 0.05) among treatments within the same parameter.

**Figure 6 cells-11-01401-f006:**
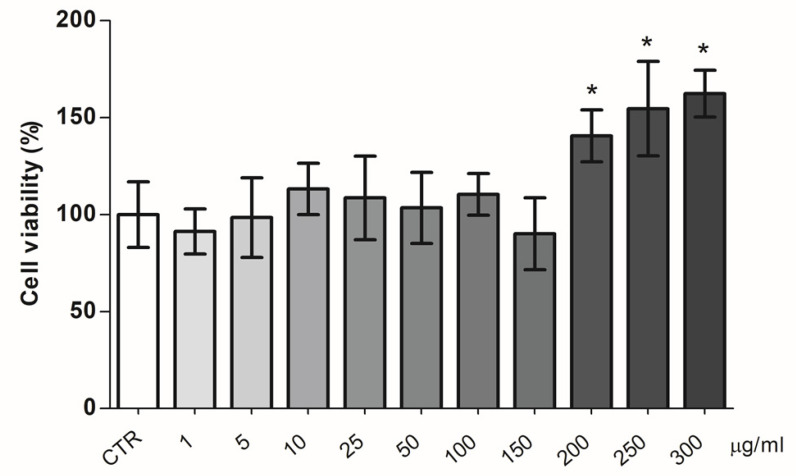
BPF effect on pAECs’ viability. Cells were treated with different doses of BPF for 5 h. * *p* < 0.05, one-way ANOVA, and post hoc Tukey’s test between each treatment vs. control (CTR) group.

## Data Availability

Not applicable.

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
