# Peer review of "Mitochondria Bioenergetic Functions and Cell Metabolism Are Modulated by the Bergamot Polyphenolic Fraction"

_cells, 2022, doi:10.3390/cells11091401_

Round 1
Reviewer 1 Report
In this study authors aimed to demonstrate the good effect of the molecules mixture of the bergamot polyphenolic fraction (BPF) on mitochondrial bioenergetic parameters including F1FO-ATPase activity, oxidative phosphorylation, mPTP desensitization to Ca2+, and finally cellular metabolism and viability in porcine Aortic Endothelial Cells. The results obtained proved the beneficial effects of BPF on mitochondrial bioenergetics and oxidative metabolism counting them as further biological targets.
In general, the subject of this study is interesting and The study design behind this research article sounds great. The methodology associated is perfect. The statistical work is sound. I will therefore have a favorable opinion for the publication of this article in Cells after careful English edit. I recommend the authors ask a native English speaking colleague to edit their paper.
Author Response
We are pleased that the Reviewer considers our work of value. We have used the language editing services of MDPI. You can find attached the certificate.

Reviewer 2 Report
The manuscript by Algieri et al. titled “The mitochondria bioenergetic functions and cell metabolism are modulated by bergamot polyphenolic fraction” details a study on the bioenergetic effects of compounds contained in the bergamot polyphenolic fraction (BPF). This fraction is rich in antioxidant and anti-inflammatory compounds which has effects that range from lowering plasma lipids to modulating sugar metabolism. They perform isolated enzyme assays, isolated mitochondrial respiratory assays, and whole cell bioenergetic assays to ascertain the benefits of BPF. There are several areas of concern that I have with this study. And while some beneficial effects were observed, their mechanistic explanation as to why is insufficient or not in line with known aspects of mitochondrial physiology. As such, I cannot endorse the manuscript for publication. My detailed critique is below.
While the authors indicate that they performed the mitochondrial isolation as quick as possible, isolating mitochondria from excised large animal hearts after slaughter results in mitochondria of poor quality. Unless the heart was immediately perfused with cardioplegia solution, the overall mitochondrial quality will rapidly degrade by the time it is cut out of the animal and handed to the laboratory members. This does not automatically disqualify studies using mitochondria from such a source; however, it certainly limits the interpretability of the data. I strongly advise that the authors confirm their results with fresh isolated mitochondria to ensure what they see is not an artifact of the dying tissue. At the very least, the time it took to extract the heart and pack it with ice should be disclosed.
When homogenizing tissue to isolate mitochondria, was the BSA used in the isolation buffer fatty-acid free?
In section 2.4, “concentration of inorganic phosphate (Pi) hydrolyzed by known amounts of mitochondrial protein in the supernatant” does not make sense. Do the authors mean ATP is hydrolyzed?
In section 2.5, I understand that the mitochondrial faction is used right after isolation and not after storage in liquid nitrogen. Is this correct? In any case, this part should be cleared up so others can understand what was done.
What does thermostatation mean? It’s not in any dictionaries that I use. I’m assuming they mean the temperature was maintained at 37 degrees Celsius.
In section 2.6, what is meant by “20 mM HEPES, pH 7.2 with TRIS?” Both HEPES and TRIS are pH buffers. If TRIS was also a reagent in the assay buffer, what was the concentration?
For those unfamiliar with the Seahorse platform, the authors should briefly explain the assays for their ATP Rate Assay, Cell Mito Stress Test, and Glycolysis Stress Test. Otherwise, the reader may misinterpret the data.
Individual data points should also be plotted with averages and standard deviations.
In Fig 1, there seems to be little to no effect at BPF concentrations below 10 ug/ml with 10-15% effects at 50-100 ug/ml. Is this effect still present at concentrations higher that 100 ug/ml. The reduction in ATPase activity when Mg2+ is replaced by Ca2+ is simply explained by the fact that the enzyme is more efficient when catalyzing reactions with the MgATP2- or MgADP- bound complexes relative to their calcium bound forms.
Based on Fig 2, the expected poor quality of the mitochondria is evident. Fresh pig heart mitochondria should yield RCR values well above 10. Also, what were the state 2 values? In the presence of Mg2+, state 4 respiration is highly variable and depends on the quality of the mitochondrial prep. This is because isolated mitochondria preps contain various levels ATPases which require Mg2+ for activity. The poorer the quality, the higher the ATPase activity. Thus, depending on the quality of the prep, the RCR data as collected in this study is not straightforward to interpret.
I also had a hard time following what the authors meant by the different letters when running the statistical comparisions. This needs to be clearer.
The experiment in Fig 3 should be repeated with the number of boluses before calcium release as the main end point. As the manuscript is currently written, I only see representative traces.
In Fig 4, how were the oxphos and glycolysis ATP production rates determined? To estimate the oxphos ATP production rate from the respiratory data, one must know the P/O ratio which depends on the type of substrate utilized by the mitochondria. I’m assuming the glycolytic ATP production rate is proportional to the ECAR value, but this isn’t well-described.
In Fig 5, BPF shows a rather dramatic effect on OCR. It appears that the OCR profile is translated down by a constant of approximately 20 pmol/min. Is this effect proportional to the BPF concentration? It appears BPF may be interfering with the Seahorse OCR measurements. Otherwise, how does the absence of BPF lead to non-mitochondrial OCR that is on par with basal OCR of BPF treated cells? In Fig 4C, why was oligomycin added before the signal stabilized? This will result in an overestimation of the glycolytic reserve. Also, OCR and ECAR should be normalized to cell number, cell mass, or mitochondrial mass.
What do the authors intend when they state, “The Ca2+-activated F1Fo-ATPase activity is documented to match the mPTP formation and opening”? This is an ambiguous statement.
The conceptualization of how BPF improves oxidative metabolism is not clear enough to follow. The authors toss out several ideas but do not link them sufficiently to ascertain their likelihood or feasibility. Based on the data presented, I do not think we are closer to understand how BPF exerts its protective effects on human health. And considering the significant antioxidant activity of BPF, mitochondrial H2O2 metabolism should have also been explored in this study.
